# On predictors of misconceptions about educational topics: A case of topic specificity

**Jana Asberger** *, Eva Thomm, Johannes Bauer

Faculty of Education, University of Erfurt, Erfurt, Germany

* jana.asberger@uni-erfurt.de

## Abstract

A large variety of misconceptions about learning, teaching, and other educational topics is prevalent in the public but also among educational professionals. Such misconceptions may lead to ill-advised judgments and actions in private life, professional practice, and policy-making. Developing effective correction strategies for these misconceptions hinges on a better understanding of the factors that make individuals susceptible to or resilient against misconceptions. The present study surveyed students from educational and non-educational fields of study to investigate whether the endorsement of four typical educational misconceptions can be predicted by study-related variables (i.e., field of study and study progress) and by students' cognitive ability (i.e., numeracy), epistemic orientations, general world views (i.e., conservative orientation), and education-related values (i.e., educational goals). A sample of $N = 315$ undergraduates in teacher education and education- and non-education-related fields of study completed an online survey. Results from structural equation models showed that the pattern of effects strongly varied across the specific misconceptions. The two misconceptions related to teaching factors (i.e., class size and effectiveness of direct instruction as a teaching method) were the most strongly affected by the field of study and had an association with conservative orientation. In contrast, the misconception about the effectiveness of grade retention as an educational intervention was more prevalent among the students emphasizing conventional educational goals, such as discipline. None of the investigated explanatory variables proved predictive of the misconception about the "feminization" of education as an educational-equity topic. Moreover, neither numeracy nor epistemic orientation was found to have any effect on the endorsement of educational misconceptions. These findings emphasize the topic dependency of the factors that make individuals susceptible to misconceptions. Future research and intervention approaches need to consider the topic specificity of educational misconceptions.

## Introduction

Both education practitioners and the general public hold a huge variety of misconceptions about learning and education, with topics ranging broadly across matters of learning, teaching, and educational policy [1–5]. Harboring such misconceptions may harm judgments and

**Data Availability Statement:** All data and analyses are available at OSF: https://osf.io/pnj6k/.

**Funding:** This research was conducted using regular budget funds from the University of Erfurt,

Faculty of Education, allocated to Johannes Bauer's chair.

**Competing interests:** The authors have declared that no competing interests exist.

actions in the context of private life, professional practice, and policymaking. Therefore, uncovering and debunking education myths [6–9] and developing interventions to initiate their correction [2, 10] have increasingly become hot topics, especially with a focus on preservice teachers. Remedying misconceptions, however, has been proven to be quite difficult [5, 11]. Developing effective interventions for such hinges not only on their design but also on deepening our knowledge about the factors that make individuals susceptible to or resilient against misconceptions and that may moderate the intervention effects [12, 13]. Previous studies with undergraduates have focused on study-related factors such as grade level or subject-related courses [4, 14]. Moreover, cognitive factors such as general cognitive ability or prior knowledge seem important, although the results on their impact are mixed [15, 16]. The focus on cold cognition, however, neglects the intrinsic societal, political, and value-laden nature of educational topics. Similar to other socio-scientific issues [17], educational topics are the subject of public debates and relate to individuals' personal interests, world views, (epistemic) orientations, and values [15, 18, 19].

The present study aimed to extend the prior research by investigating a broader array of predictors of misconceptions about educational topics with undergraduates from various fields of study. Specifically, we first investigated how study-related variables (i.e., field of study and study progress), students' cognitive ability (i.e., numeracy), epistemic orientations, general world views (i.e., conservative orientation), and education-related values (i.e., educational goals) predict students' endorsement of typical educational misconceptions. Second, we were interested in the extent to which these effects are topic-specific or occur consistently across misconceptions on different educational topics. Such topic specificity could at least partially explain the difficulty of finding effective interventions against misconceptions. When misconceptions are shaped by different individual factors, these factors may also influence the effectiveness of refuting them. Drawing upon the work of Asberger et al. [1], we focused on misconceptions about four typical education-related topics: (a) class size effects; (b) grade retention effects; (c) effectiveness of direct instruction; and (d) effects of "feminization" in elementary and primary education on boys' development. These topics have frequently been the subject of public discussion and touch on prominent categories of educational topics, such as teaching (methods) in class, educational interventions, and equity issues.

This study extends prior research (both theoretically and empirically) by examining individual differences in misconceptions about educational topics. Considering the socio-scientific nature of educational topics, we specifically shed light on the relationship between misconceptions and individual epistemic orientations, worldviews, and values–factors which, despite their importance, have not been considered before. Further, this study has practical implications for the development of interventions to correct misconceptions, especially in the context of higher education. Understanding predictors of educational misconceptions can help to facilitate the teaching of research-based educational knowledge that contradicts students' prior beliefs; this may be particularly important in teacher education. In the next sections, we will review the relevant studies on educational misconceptions and their consequences before discussing their potential predictors.

## Misconceptions in education

Misconceptions are generally defined as individual beliefs that partly or completely contradict the currently accepted body of research on a given topic [3, 4, 20]. Research has identified a great variety of misconceptions in education (our literature review identified more than 60), some of which are widely believed by the public [6–9, 21] for several reasons. First, everybody develops a set of beliefs about education from his or her own experiences and from anecdotal

experiences of others conveyed though narratives or the media [11]. The immediacy of such first- or second-hand experiences can provide subjectively sufficient evidence to justify such beliefs, obliterating the need for external verification [22, 23]. Second, assumptions about educational topics strongly relate to personal attitudes and values [18, 19], making them even more challenging to refute [24]. Third, personal views of what makes knowledge justified (i.e., epistemic orientation) may enforce the above processes when individuals consider their personal experience or intuitive persuasiveness more important than scientific evidence [11, 25]. Finally, even when one is confronted with research-based evidence, the probabilistic and preliminary nature of scientific knowledge especially in the so-called soft sciences like education [26, 27] may lead one to discount and reject research in favor of one's own beliefs [5]. When individuals' prior beliefs contradict research evidence, they tend to re-interpret, devalue, or even ignore the evidence altogether [28–30], and may even dismiss the potency of research to provide any relevant knowledge [23, 31].

Although it is not the focus of this study, it is worth noting that misconceptions can have tangible negative consequences. In the case of teachers, for example, it has been shown that educational misconceptions can negatively affect their teaching quality [22, 32–34]. Therefore, addressing educational misconceptions is a particular challenge for a research-oriented teacher education program and for the efforts to make education more evidence-based. Hence, understanding the driving factors of educational misconceptions and providing effective interventions for such is important for informed discussions and decisions not only in private or civic life but also for improving professional practice [32, 35, 36].

**Examining individual differences in the endorsement of educational misconceptions.**
Although numerous studies have focused on the predictors of educational misconceptions, there is currently no theoretical model that explains the individual differences in the endorsement of misconceptions. The previous studies on the predictors of educational misconceptions have focused on cognitive traits such as general cognitive ability, critical thinking, or subject-related knowledge, and for the studies that investigated university students, study-related variables [3, 14, 16]. In the following sections, we will elaborate on these variables before turning to epistemic orientations, general worldviews, and education-related values as further potential influencing factors. We will also draw from the research on misconceptions about psychology (e.g., about memory, child development, or brain functioning) that are relevant to our purpose.

**Study-related characteristics and aspects of cognitive ability.**  Study-related characteristics such as the subject-related courses attended, study progress, academic performance, and having a graduate degree have been found to reduce misconceptions in a number of studies [1, 4, 14, 20, 37]. One of the presumed mechanisms behind these effects is that university education exposes students to research-based knowledge, thereby helping to correct their misconceptions. In that sense, such study-related characteristics can be seen as proxy variables for domain-relevant knowledge. In the present study, we investigated the role of *field of study*, *study progress*, and for student teachers, *school-based experiences* from internships. Specifically, we analyzed the differences among students enrolled in teacher education, those enrolled in other education-related fields of study (e.g., educational science, psychology), and those enrolled in non-education-related fields of study (e.g., economy) and determined if study progress has differential effects on educational misconceptions. At first glance, teacher education and other education-related fields appeared similar in addressing educational misconceptions, but teacher education has a more specific focus on topics directly related to learning and instruction in the classroom. Hence, we assumed that preservice teachers endorse classroom-related misconceptions less strongly [1, 14]. For example, because direct instruction is often taught as an effective method in teacher education, we assumed that preservice teachers would

less strongly endorse the notorious misconception that it is inferior to the "active" forms of instruction [9]. However, preservice teachers may also be prone to endorse specific misconceptions that are consistent with the practice in schools. For example, they may (falsely [38]) believe that grade retention is an effective educational intervention simply because it is actively being practiced in many countries [39]. Hence, the effect of field of study may be topic-dependent [1]. Finally, school-based internships that are a compulsory part of teacher education in many countries [40, 41] may have a double-edged effect on misconceptions. On the one hand, they may enhance professional knowledge and thereby reduce educational misconceptions [42]; on the other hand, the subjective nature of personal experiences makes them prone to confirmation bias and may thus reinforce educational misconceptions [6, 43].

Beyond these factors, general cognitive ability is crucial for understanding scientific information and identifying the faulty reasoning underlying many misconceptions [4, 20]. Specifically, evidence suggests that a lack of *numeracy* (i.e., individuals' ability of probabilistic and statistical reasoning [44]) is associated with endorsement of questionable beliefs and susceptibility to misinformation [45–47]. Numeracy is also important because understanding the results of empirical research requires probabilistic interpretation [27, 48, 49]. For example, appreciating that an educational intervention can be effective on average even if its effect does not apply equally to all students requires an understanding of probabilistic causality and distribution. Therefore, we assumed that less numerate students more strongly tend to endorse educational misconceptions.

**Epistemic orientations, world views, and educational values.** Endorsement of misconceptions may also relate to more subjective personal orientations, worldviews, and values. First, there is evidence that dysfunctional *epistemic orientations* (i.e., personal views on how to determine what is true) make individuals susceptible to questionable beliefs [15, 25]. For example, Garrett and Weeks [25] found that people who believe that truth can be identified by intuition (*faith in intuition*) and is strongly determined by social and political negotiation (*truth is political*) are more likely to adhere to conspiracy theories. In contrast, the belief that truth depends on justification by external evidence (*need for evidence*) was found to be a protective factor. In the present study, we tested whether analogous effects occur for the endorsement of misconceptions in education. For example, we investigated if persons who are convinced that personal insights are sufficient for warranting statements about educational topics may be more inclined to accept information that is scientifically invalid [15, 50]. Likewise, belief in the political determination of the truth may cause bias in the processing of information about educational topics to serve one's political orientation (see below) or a high need for evidence may mitigate the risk of adopting unproven claims and may increase one's willingness to change one's misconceptions when faced with corrective facts.

Second, beliefs about education need to be considered along with the person's worldviews and values [18, 19]. Political orientations, values, and stances can have a direct impact on education and on one's beliefs about it. For example, the grade retention of low-performing students is more strongly preferred by conservatives [51]. In addition, teacher-centered methods like direct instruction are often considered conservative compared to the supposedly more progressive constructivist methods in public and scientific debates [52]. Political orientation can also cause bias in information processing in favor of personal beliefs [28, 30, 53]. For example, Garrett and Weeks [25] found that a conservative orientation is positively related to dysfunctional epistemic orientations (i.e., faith in intuition, truth is political) and leads to less accurate judgments about socio-scientific and political issues. Similarly, there is evidence of an association between a conservative orientation on the one hand and susceptibility to misinformation and motivated rejection of science regarding socio-scientific issues on the other [47, 54]. There is currently no evidence that misconceptions about educational topics are also

related to political orientation, but we expected preservice teachers who consider themselves more conservative than liberal (*conservative orientation*) to view educational topics in line with their political ideology. Hence, we assumed them to have stronger misconceptions about grade retention but lower misconceptions about direct instruction. We did not have specific hypotheses about the other investigated topics but nonetheless explored their potential effects.

Next to general political orientations, we considered more education-specific values in terms of *educational goals*. Educational goals represent the core values underlying education on the basis of the importance that one assigns to the various objectives of children's cognitive, affective, and social development [55, 56]. The typical categories are intellectual goals (e.g., acquiring broad knowledge in diverse fields), social goals (e.g., developing social competence), and the "conventional goals" (i.e., the classical virtues regarding achievement and discipline [57]). Similar to the above reasoning on conservative orientation, we assumed the association of the conventional and intellectual goals with misconceptions about direct instruction and grade retention. We did not have specific hypotheses about social goals, but we were interested in exploring their potential effects.

## The present study

The goal of this study was to investigate whether and how student characteristics relate to questionable beliefs in education. In particular, we addressed the following research questions: (1) How do (i) study-related characteristics and aspects of cognitive ability, (ii) epistemic orientations, and (iii) conservative orientation and educational goals predict students' endorsement of typical educational misconceptions? And (2) are these effects topic-specific?

In the previous section, we elaborated on the nature of potential relationships between misconceptions and the different individual factors from prior literature. Table 1 provides a summary of our assumptions. Before collecting data, we preregistered all hypotheses, study design, measures and analytic procedures (http://aspredicted.org/blind.php?x=8375mh). As aforementioned, some effects were tested exploratory when prior knowledge was too scarce to specify a clear assumption, which was a concern encountered most frequently with topic-specific variations of the effects.

## Methods

### Data collection and participants

We preregistered an effective sample size of 300 to have sufficient statistical power for the planned analyses. Considering the drop-out rate associated with conducting online surveys, we aimed at recruiting about 450 participants to compensate for the expected dropouts and exclusions during the data cleaning. Students of different disciplines in German universities were recruited via mailing lists and the social media, and wherever possible, offline, via classroom lectures. Vouchers were given away through a lottery as incentives for participation in the study. The University of Erfurt's Internal Review Board (#20210517) confirmed that the study was a negligible-risk research without a foreseeable risk of harm or discomfort on the part of the participants. We followed the American Psychology Association (APA) [58] ethical principles of psychologists, including voluntary participation based on informed consent.

Overall, 422 participants who gave their informed consent to participate in the study responded to the online survey. We excluded data from the following, according to the preregistered criteria: from the 16 participants who indicated that they were not students; from the 37 participants who completed the questionnaire within an unreasonably short time (i.e., in less than 10 minutes); and from 54 multivariate outlier cases (minimum covariance determinant, $p \leq .01$ [59]). We replaced the probable and extreme univariate outlier values by missing

**Table 1. Summary of the preregistered hypotheses.**

|  | Misconceptions about… | | | |
|---|---|---|---|---|
|  | **Class size** | **Grade retention** | **Direct instruction** | **Feminization** |
| *Study-related characteristics* |  |  |  |  |
| Field of study[a] |  |  |  |  |
| Education-related | + | - | 0 | 0 |
| Non-education-related | + | 0 | + | 0 |
| Study progress[b] | - | - | - | - |
| School-based experience[c] | ? | ? | ? | ? |
| *Cognitive ability* |  |  |  |  |
| Numeracy | - | - | - | - |
| *Epistemic orientations* |  |  |  |  |
| Faith in intuition | + | + | + | + |
| Need for evidence | - | - | - | - |
| Truth is political | + | + | + | + |
| *World views and values* |  |  |  |  |
| Conservative orientation | ? | + | - | ? |
| *Educational goals* |  |  |  |  |
| Intellectual | ? | + | - | ? |
| Social | ? | ? | ? | ? |
| Conventional | ? | + | - | ? |

[a] Teacher education as the reference group.

[b] For preservice teachers and students from education-related studies.

[c] For preservice teachers.

+ Positive correlation;—negative correlation; 0 correlation close to zero;? exploratory.

values (probable outlier $z > 2.58$; extreme outlier $z > 3.29$). Although the number of excluded participants may seem substantial, this is common in online surveys and is done to ensure data quality.

To test the potential biases through the exclusion procedure, all the analyses were conducted twice, once including the identified outliers and once excluding them. Only small differences occurred, which did not substantially alter the conclusions. Consistent with the preregistration for the analytic procedures, we will report the results herein excluding the outliers. The results including the outliers are available in the electronic supplement (S1 Table).

The analyzed sample consisted of 315 undergraduates. The participants indicated whether they were enrolled in a teacher training program (32.38%), other education-related study program (e.g., psychology or educational science; 40.00%), or non-education-related study program (e.g., economics or natural sciences; 18.09%; 9.52% did not indicate the study program in which they were enrolled).

## Measures

**Misconceptions.** The participants' educational misconceptions were measured by the Questionable Beliefs in Education Inventory (QUEBEC [1]) questionnaire. This questionnaire addresses the four aforementioned typical educational misconceptions: (a) that *class size* strongly influences teaching quality and student learning; (b) that *grade retention* is an effective intervention for low-achieving students; (c) that *direct instruction* is less effective than "active" forms of instruction; and (d) that the high proportion of female teachers in elementary and

primary education (*feminization*) explains the lower educational outcomes of boys compared to girls. These misconceptions frequently surface in public debates and in the media, and have been proven to be misconceptions by a substantial body of literature, including meta-analyses (e.g., class size [60]; grade retention [61]; direct instruction [62]; and feminization [63]). In a previous study [1], we reported that the class size misconception had the highest prevalence among the preservice teachers in the study, that the grade retention and direct instruction misconceptions had medium-level prevalence, and that the feminization misconception had a low but noticeable prevalence. Per topic, the subscales consist of four to seven items (e.g., "Grade retention of low-performing students strongly remedies their knowledge deficits"). The participants rated the extent to which they believed the given statements were correct on a 6-point Likert scale. The order of the topics and the order of the items within each topic were randomized across the participants.

Because we wanted to use structural equation modeling (SEM) to test our hypotheses, we ran preparatory confirmatory factor analyses (CFA) and invariance tests to ensure a sound measurement model of the misconceptions [64]. The results showed an acceptable model fit for the CFA after excluding three items that had been previously proven to be problematic [1]: $\chi^2$ (129) = 192.014; p < .001; root mean square error of approximation (RMSEA) = .039; comparative fit index (CFI) = .965; and standardized root mean residual (SRMR) = .044. Scalar invariance held for the comparison of the students with different fields of study (S2 Table).

**Predictors.** The student participants in this study reported their *field of study* (teacher education program; education-related program; non-education-related program) and *study progress* (semester). The preservice teachers additionally answered how many weeks of school-based internship they had completed (*school-based experience*). *Numeracy* was tested by the Berlin Numeracy Test [44], which contains four open-ended questions regarding probabilistic problems (e.g., "Imagine we are throwing a five-sided die 50 times. On average, out of these 50 throws, how many times would this five-sided die show an odd number (1, 3, or 5)?"). Epistemic orientations were measured by three scales [25] on *faith in intuition* (e.g., "I trust my gut to tell me what's true and what's not"), *need for evidence* (e.g., "Evidence is more important than whether something feels true"), and *belief that truth is political* (e.g., "Facts are dictated by those in power"). The participants indicated their agreement with each statement on a Likert scale ranging from 1 (= "strongly disagree") to 9 (= "strongly agree"). *Conservative orientation* was captured using a single item asking the students to assess their own orientation on a Likert scale ranging from 1 (= "extremely liberal") to 10 (= "extremely conservative") [65]. Finally, the students answered subscales on three categories of educational goals [66]: conventional goals (e.g., "Order and discipline"); intellectual goals (e.g., "Solid knowledge in major subjects"); and social competence goals (e.g., "Adequate social behavior"). The participants were instructed to assess the extent to which they considered each goal important in school on a Likert scale ranging from 1 (= "less important") to 4 (= "extremely important").

## Analyses

For the SEM analyses, we used the lavaan package (Version 0.6–7) in R and robust full information maximum likelihood estimation to account for non-normality and missing data. The model fit was judged on the basis of standard goodness-of-fit indices and common cut-off values: CFI (good: 1–.95; acceptable: .95–.90), RMSEA (good: 0–.05; acceptable: .05–.08), and SRMR (good: 0–.05; acceptable: .05–.10). To reduce the model complexity, we estimated separate models for the four misconceptions. The multi-item subscales on epistemic orientations and educational goals entered the model as aggregated mean scores, and we controlled for measurement error using the single-indicator latent variable approach [67], fixing the

respective residual terms on the basis of the scale reliability estimates. The same was done for numeracy, but on the basis of the estimated person parameters from a two-parameter logistic item response theory model [68]. To test the effect of school-based experience, we analyzed the four models using the subsample of preservice teachers, adding this predictor to the model.

## Results

### Descriptive statistics

The descriptive statistics and reliability values (McDonald's Ω) are presented in Table 2. Regarding the four misconceptions, the students showed highest agreement with those about class size and direct instruction. The endorsement of misconceptions about grade retention was also above midpoint in the answer scale. In contrast, the misconceptions about feminization were less prevalent. This pattern resembles the average prevalence values previously reported [1]. All the scales, except those for the intellectual goals, had acceptable reliability values. For completeness, we report the results on the intellectual goals herein, but we shall not interpret them. The reliability values for the direct-instruction misconception and the

**Table 2. Descriptive statistics of the educational misconceptions and predictors.**

| | *N* | # Items | *M* | *CI* | *SD* | Ω |
|---|---|---|---|---|---|---|
| *Misconceptions*[a] | | | | | | |
| Class size | 306 | 4 | 4.77 | [4.67; 4.87] | 0.89 | .75 |
| Grade retention | 303 | 7 | 3.39 | [3.30; 3.48] | 0.82 | .78 |
| Direct instruction | 306 | 5 | 4.18 | [4.09; 4.28] | 0.83 | .66 |
| Feminization | 304 | 4 | 1.86 | [1.77; 1.96] | 0.82 | .90 |
| *Study-related characteristics* | | | | | | |
| Study progress[b] | 312 | 1 | 4.84 | [4.47; 5.21] | 3.34 | - |
| School-based experience[c] | 99 | 1 | 13.73 | [9.46; 18.00] | 21.68 | - |
| *Cognitive ability* | | | | | | |
| Numeracy[d] | 274 | 4 | 1.61 | [1.47; 1.75] | 1.22 | .81 |
| *Epistemic orientations*[e] | | | | | | |
| Trust in intuition | 285 | 4 | 5.43 | [5.28; 5.59] | 1.33 | .77 |
| Need for evidence | 285 | 4 | 6.27 | [6.09; 6.44] | 1.52 | .80 |
| Truth is political | 285 | 4 | 4.50 | [4.30; 4.70] | 1.71 | .77 |
| *World views and values* | | | | | | |
| Conservative orientation[f] | 282 | 1 | 3.87 | [3.68; 4.06] | 1.61 | - |
| Educational goals[g] | | | | | | |
| Intellectual goals | 292 | 6 | 3.02 | [2.97; 3.07] | 0.46 | .52 |
| Social goals | 292 | 8 | 3.20 | [3.15; 3.26] | 0.44 | .71 |
| Conventional goals | 292 | 3 | 2.93 | [2.86; 2.99] | 0.57 | .65 |

[a] Six-point Likert scale (*do not agree at all* [1] to *fully agree* [6]).

[b] Semesters of study.

[c] Number of weeks (subsample of preservice teachers only).

[d] Number of correct answers (0–4).

[e] Nine-point Likert scale (*do not agree at all* [1] to *fully agree* [9]).

[f] Ten-point Likert scale (*extremely liberal* [1] to *extremely conservative* [10]).

[g] Four-point Likert scale (*less important* [1] to *very important* [4]); the number of response categories varied because we used the instruments' original answer formats, respectively.

Ω = McDonald's Ω reliability.

conventional goals were somewhat below .70 but were comparable to the values obtained by other studies using these scales [1, 57].

## Predicting misconceptions in education

The goodness of fit for the four analyzed models was acceptable, with the exception of the CFI for the model of direct instruction (S3 Table). Given that the other goodness-of-fit indices were within an acceptable to good range for this model, however, we decided to proceed with it. The standardized regression estimates for all the models are listed in Table 3.

The effect of *field of study* was topic-specific. As expected, the preservice teachers agreed less strongly than the students of other fields of study with the misconceptions about class size and direct instruction. However, field of study did not predict the misconceptions about grade retention and feminization. Fig 1 illustrates the group differences for all the four topics. *Study progress* was negatively related to the misconception about grade retention (small effect) while it had no effect on the other topics. To explore the potential differential effects of study progress across students of different fields of study, we added the respective interaction terms to the models (field of study x study progress). None of the interaction effects was statistically significant or of a substantial effect size, however, so we removed them from the models.

Neither *numeracy* nor *epistemic orientations* predicted any of the four educational misconceptions. The corresponding hypotheses are thus rejected.

*Conservative orientation* predicted the misconception about direct instruction in the expected direction; that is, more conservative students agreed less strongly with it. The effect size was medium. Contrary to our assumption, however, more conservative students also agreed less strongly with the misconception about class size. The effect size was small.

**Table 3. Standardized regression estimates from the structural equation models of educational misconceptions.**

| | Class size | | Grade retention | | Direct instruction | | Feminization | |
|---|---|---|---|---|---|---|---|---|
| Predictor | β | *p* | β | *p* | β | *p* | β | *p* |
| *Study-related characteristics* | | | | | | | | |
| Field of study[a] | | | | | | | | |
|  Education-related | **.32** | **.04** | -.04 | .79 | **.77** | **< .001** | .01 | .95 |
|  Non-education-related | **.51** | **< .01** | -.14 | .45 | **.67** | **< .001** | -.11 | .57 |
| Study progress | -.06 | .32 | **-.14** | **.04** | -.04 | .56 | .00 | .98 |
| *Cognitive ability* | | | | | | | | |
| Numeracy | .07 | .30 | .08 | .34 | -.09 | .35 | -.01 | .94 |
| *Epistemic orientations* | | | | | | | | |
| Faith in intuition | -.02 | .79 | .00 | .93 | .08 | .37 | -.04 | .66 |
| Need for evidence | .08 | .46 | .00 | .97 | -.04 | .70 | .19 | .07 |
| Truth is political | .01 | .90 | -.16 | .11 | .11 | .28 | .13 | .19 |
| *World views and values* | | | | | | | | |
| Conservative orientation | **-.18** | **.05** | .02 | .89 | **-.21** | **.04** | -.01 | .94 |
| Educational goals | | | | | | | | |
|  Intellectual | .00 | .99 | -.27 | .22 | .14 | .51 | -.24 | .28 |
|  Social | -.04 | .78 | .01 | .92 | .00 | .99 | .04 | .78 |
|  Conventional | .26 | .18 | **.50** | **.01** | -.11 | .56 | -.07 | .70 |
| $R^2$ | .12 | | .17 | | .22 | | .10 | |

Boldface = $p \leq .05$; italicized = effect as predicted.

[a] Teacher education as a reference group, estimates standardized on outcome.

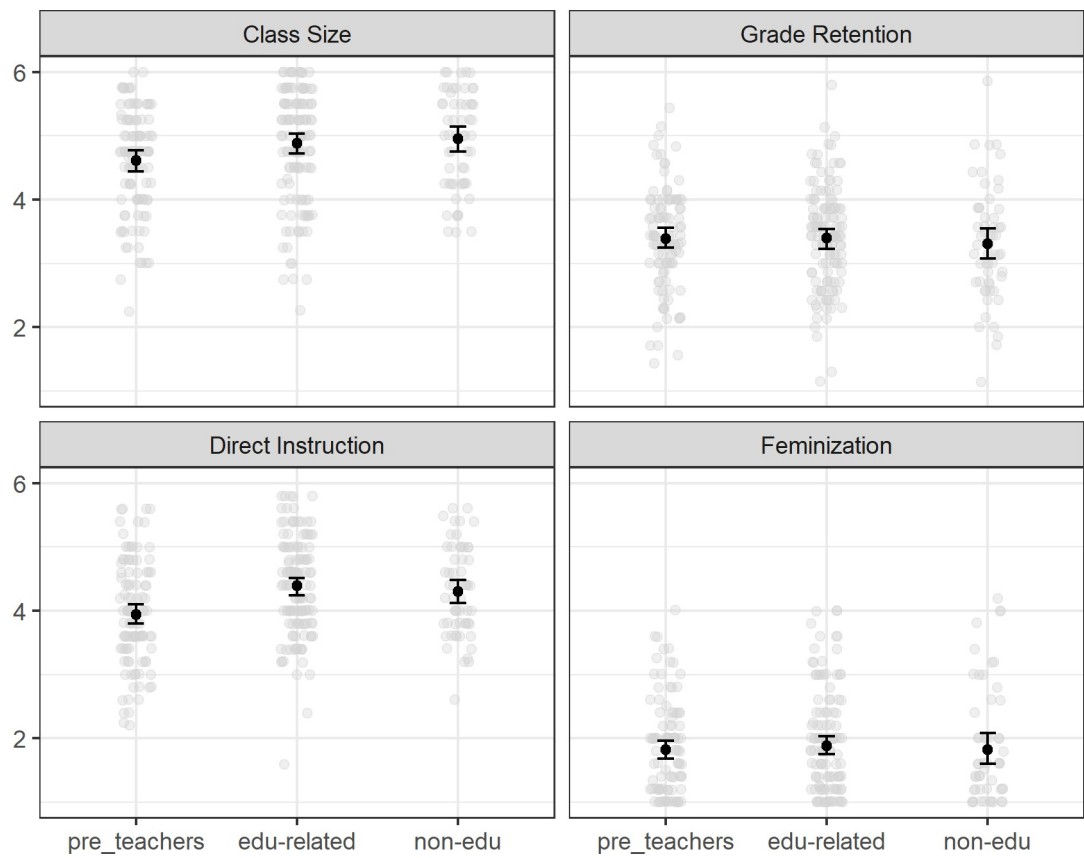

**Fig 1. Educational misconceptions by field of study.** The figure shows the means and 95% confidence intervals of the agreement to the educational misconceptions across the three student groups in this study. Higher values indicate higher agreement.

Regarding *educational goals*, only one of the assumed effects was statistically significant; that is, the students who gave more importance to conventional goals agreed more strongly with the misconception about grade retention. For social goals, the exploratory analysis delivered null effects throughout.

The results for *school-based experience* in the subsample of preservice teachers showed a small positive effect on the misconception about class size ($\beta = .19$; $p = .03$), but no further effects. That is, the future teachers who had more weeks of practical experience in school showed higher misconceptions about class size.

## Discussion

In this study, we sought to illuminate the individual characteristics that could shape misconceptions about educational topics. Particularly, we investigated if students' study-related characteristics (i.e., field of study and study progress), cognitive ability (i.e., numeracy), epistemic orientations, general world views (i.e., conservative orientation), and education-related values (i.e., educational goals) predict the typical misconceptions about educational topics. Our results showed, first, that the pattern of effects strongly varied by misconception. Second, numeracy and epistemic orientations were completely unrelated to educational misconceptions, although such associations were expected on the basis of the prior theorizing and evidence. Our findings emphasize the topic dependency of characteristics that make individuals

susceptible to misconceptions, and may thus provide new approaches for research on the prevalence and refutation of misconceptions in education.

## Predictors of educational misconceptions are topic-specific

Consistent with both the prior findings and our hypotheses, we found that study-related characteristics have effects on endorsement of educational misconceptions. As predicted, the preservice teachers endorsed misconceptions directly related to teaching in class (i.e., about class size and direct instruction) less strongly than the students of other fields of study [1, 14]. The lower prevalence of these misconceptions among the preservice teachers may indicate that preservice teachers acquire topic-relevant knowledge from their studies that corrects or prevents such misconceptions. However, we have no data on the degree to which these topics were actually addressed in the lectures. Also noteworthy is the finding that the misconceptions about class size and direct instruction had a relatively high prevalence across all the investigated groups. Thus, the preservice teachers were by no means free of them. Our findings even show that school-based experiences seem to strengthen preservice teachers' misconception about the role of class size. Although the effect was small, it is consistent with the earlier findings that preservice teachers find their first teaching experiences quite stressful [69]. This may be aggravated by the perceived large class sizes.

Moreover, unlike in our first study [1], we did not find preservice teachers more susceptible to misconceptions about grade retention. We had assumed so because preservice teachers might mistake the existing practice in schools as evidence that grade retention is an effective instrument to support struggling students. Study progress and preservice teachers' practical experiences were mostly unrelated to educational misconceptions, barring the small negative association between study progress and misconceptions about grade retention.

We also found topic-specific effects for conservative orientation and conventional goals. Regarding conservative orientation, the results corroborated the hypothesized relation with misconceptions about direct instruction, which can be seen to be in line with the traditional view of teaching [52]. We did not find the expected relationship for misconceptions about grade retention, though, but we observed a negative association between conservative orientation and misconceptions about the role of class size in our exploratory analysis. A potential reason for this could be that a conservative orientation may coincide with adherence to the traditional view that class size is a less important factor.

For conventional goals, the pattern of results was reverse: no effect on the misconception regarding direct instruction was found. However, the misconception about the effectiveness of grade retention as an educational intervention was more prevalent in the students emphasizing conventional educational goals. A similar interpretation may be employed here: grade retention may be directly related to a lack of achievement and discipline, which are at the core of the conventional educational goals [55, 57].

In summary, the aforementioned findings of this study support the view that political orientations and educational goals can indeed have an impact on specific educational misconceptions. We therefore conclude that these relationships warrant further investigation. The role of these factors may be particularly interesting in the context of misconceptions about value-laden educational topics (e.g., inclusive schooling) and related interventions.

## Educational misconceptions are not only hard to change but also hard to predict

Some of the most striking findings of this study are in fact null findings. First, we found neither effects of numeracy as an aspect of general cognitive ability nor of epistemic orientations,

which supposedly play an important role in misconceptions, scientific-information processing, and conceptual change [20, 25, 45, 47]. Second, none of the investigated variables proved predictive of the misconception about feminization in early childhood and primary education. Unlike the other misconception topics, neither study-related characteristics nor general worldviews seem related to the misconception about feminization. This may be because this educational-equity topic may be too specific. It may thus be interesting to instead examine gender differences or other sociodemographic factors as predictors of educational misconceptions.

What are potential reasons for these null results? First, one explanation could be that our study omitted important individual characteristics. As there is so far no clear theoretical model from which predictors could be derived, we mainly based our assumptions on previous research on educational misconceptions. Although we deduced the set of predictors from a broad literature review, covering both theoretical and empirical works, further individual characteristics may play a role. For example, Sinatra and Jacobson [11] suggest that dispositions toward thinking and reasoning, such as open-minded reasoning strategies, willingness to engage with new information, and enjoyment of effortful thinking, may shape misconceptions.

Second, another potential explanation is that educational misconceptions may also be shaped by non-individual factors such as the characteristics of the topic itself (e.g., emotional load), its presence in popular culture and public debate, and influences of the social context (e.g., group identity processes, social media) [70]. This could explain why some misconceptions are widespread while some are not. For example, in a teacher survey, 93% of the participants from the United Kingdom and 96% of those from the Netherlands indicated that they believe that individuals learn better when they receive information in their preferred learning style [16]. The topic of learning styles is especially emotionally loaded because it offers a way for teachers to help their students and tells learners something about themselves [11]. The commercial marketing of brain-based programs, books, and workshops reinforced the spread of the learning style myth [16]. Thus, the prevalence of this myth probably has non-individual causes, beyond the individual ones investigated in this study.

Third, it might be that our measures did not successfully capture the variables under investigation. Though we used measures that were applied successfully in earlier research, they may not have worked well in the context of the present study.

To summarize, misconceptions about educational topics appear persistent, and our results underline that they are also hard to predict. Moreover, as explained below, our findings have relevant theoretical and practical implications for future research and higher education.

## Implications for research and higher education

As previously discussed, misconceptions may occur in "clusters" that have similarities in content and structure (e.g., referring to institutional practices or teaching practices) and that maintain or reinforce each other [11]. For example, Macdonald et al. [14] identified a cluster of specific neuromyths bearing similar characteristics that could be addressed simultaneously in interventions. Future research should examine commonalities in the structure and characteristics of common topics of misconceptions. Attaining a deeper understanding of what makes an educational topic prone to misconceptions will provide an important additional perspective to address those misconceptions. If such commonalities can be identified, they may also contribute to a better understanding of the topic-specific relations found in this study and may help explain the persistence of educational misconceptions.

Furthermore, considering the socio-scientific nature of educational matters, the impact of personal views and stances needs to be considered more thoroughly. Existing intervention

approaches, such as refutational texts [2, 10], predominantly aim to change individuals' beliefs by replacing dysfunctional explanations and providing accurate knowledge. However, these studies pay only little attention to individuals' attitudes and values, which may need to be addressed in addition to cognitive factors in order to overcome misconceptions. This may be a reason why the findings on the impact of such interventions are mixed. Only recently, Thacker et al. [24] shed light on attitudes and values as influencing factors, inspecting the effects of measures they called "persuasive refutation texts". Our results support the importance of further developing such approaches to mitigate questionable beliefs, and this may play a specific role in interventions in higher education and teacher education. In particular, preservice teachers who have been found to belief in a number of educational myths when entering teacher education programs may profit from lectures with successful debunking strategies [5]. Interventions that take into account a recipient's personal views and stances can be promising for educating future professionals towards more evidence-based decisions and actions.

## Study limitations

This study had several limitations that should be noted. First, it focused on undergraduates because we were interested in the role of educational misconceptions in knowledge acquisition in higher education. Hence, it is unclear if and to what degree our findings can be generalized to samples beyond university students, such as experienced teachers and the general public. Second, the cross-sectional design of this study prohibits drawing causal inferences from the study, and any causal assumptions implied in our hypotheses were based only on theoretical considerations and prior research. Third, although the preregistered hypotheses could be tested in a confirmatory way, a substantial part of this study was exploratory. Thus, although the study findings can inform further research, they should be interpreted cautiously for the time being. Fourth, this study addressed only a small selection of educational misconceptions. We focused on typical educational misconceptions that surface widely in public debates at least in Germany. However, this approach enabled us to capture various aspects of each misconception in multiple items, and to provide reliability estimates for them. In contrast, the earlier research frequently employed single-item measures [3, 5, 15]. Nevertheless, for advancing our knowledge about the topic specificity of educational misconceptions, it is important to investigate more educational misconceptions than those considered here. Despite these limitations, we believe that the present study offers new perspectives for future research on educational misconceptions and for developing evidence-based interventions to address them in higher education.

## Supporting information

**S1 Table. Results with outliers.**
(DOCX)

**S2 Table. Results of the measurement invariance analysis.**
(DOCX)

**S3 Table. Goodness-of-fit indices for the structural equation models.**
(DOCX)

## Author Contributions

**Conceptualization:** Jana Asberger, Eva Thomm, Johannes Bauer.

**Data curation:** Jana Asberger.

**Formal analysis:** Jana Asberger.

**Investigation:** Jana Asberger.

**Methodology:** Jana Asberger, Eva Thomm, Johannes Bauer.

**Project administration:** Jana Asberger.

**Supervision:** Eva Thomm, Johannes Bauer.

**Validation:** Johannes Bauer.

**Visualization:** Jana Asberger.

**Writing – original draft:** Jana Asberger.

**Writing – review & editing:** Jana Asberger, Eva Thomm, Johannes Bauer.

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
