## [Decision Letter · Decision Letter 0]

24 Aug 2021

PONE-D-21-14744

On predictors of misconceptions about educational topics: A case of topic specificity

PLOS ONE

Dear Dr. Asberger,

Thank you for submitting your manuscript to PLOS ONE. After careful consideration, we feel that it has merit but does not fully meet PLOS ONE’s publication criteria as it currently stands. Therefore, we invite you to submit a revised version of the manuscript that addresses the points raised during the review process.

Please could you expand on the conclusions drawn from your results to put the work into context and to explain its impact.

We look forward to receiving your revised manuscript.

Kind regards,

Andrew R. Dalby, PhD

Academic Editor

PLOS ONE

Journal Requirements:

Reviewers' comments:

Reviewer's Responses to Questions

**Comments to the Author**

1. Is the manuscript technically sound, and do the data support the conclusions?

Reviewer #1: Partly

2. Has the statistical analysis been performed appropriately and rigorously? 

Reviewer #1: Yes

3. Have the authors made all data underlying the findings in their manuscript fully available?

Reviewer #1: Yes

4. Is the manuscript presented in an intelligible fashion and written in standard English?

Reviewer #1: Yes

5. Review Comments to the Author

Reviewer #1: Comments and Suggestions for Authors

I appreciated the manuscript which focused on a significant topic and appeared to be clearly organised. The sections corresponding to Introduction and Background (from Introduction to Epistemic orientations, world views, and educational values) seemed to be adequate overall. Conversely, the research questions were not explicity given. Authors should explicity list them in order to favour the readers’ comprehension of the results and the conclusions. From this point of view, the sentence “The present study aimed to test the study hypotheses about the predictors of the educational misconceptions mentioned earlier” (p.10) was too vague. Moreover, the goal was not specified explicitly enough. Authors should explain why they prepared the paper and who should profit from the new knowledge acquired in their investigations. Unfortunately, authors failed to specify what type of benefits may result from this study and what type of stakeholders may be interested in it in terms of policy decisions.

I would like to highlight the satisfactory description of the research design. Methods were adequately described. The results discussion had merits, but it was a bit thin in interpretations. For instance, the authors claimed: “Some of the most striking findings of this study are in fact null findings…..(p. 21, line 439). No comment? Why? Consequences?

6. PLOS authors have the option to publish the peer review history of their article (what does this mean?). If published, this will include your full peer review and any attached files.

Reviewer #1: No

---

## [Author Response · Author response to Decision Letter 0]

26 Oct 2021

Dear Reviewer,

We thank the editor and the reviewer very much for their helpful comments and recommendations regarding our manuscript; we believe the manuscript has profited substantially from them.

We provide a detailed account of the revisions we have made in the attachment. 

We hope we have addressed the reviewer’s and editor’s concerns adequately and that the reviewer judges the revised article to be suitable for publication in PLOS ONE.

---

## [Editor Report · Decision Letter 1]

29 Oct 2021

On predictors of misconceptions about educational topics: A case of topic specificity

PONE-D-21-14744R1

Dear Dr. Asberger,

We’re pleased to inform you that your manuscript has been judged scientifically suitable for publication and will be formally accepted for publication once it meets all outstanding technical requirements.

Kind regards,

Andrew R. Dalby, PhD

Academic Editor

PLOS ONE
---

## [Editor Report · Acceptance letter]

19 Nov 2021

PONE-D-21-14744R1 

On predictors of misconceptions about educational topics: A case of topic specificity 

Dear Dr. Asberger:

I'm pleased to inform you that your manuscript has been deemed suitable for publication in PLOS ONE. Congratulations! Your manuscript is now with our production department. 

Kind regards, 

on behalf of

Dr. Andrew R. Dalby 

Academic Editor

PLOS ONE